# Antitumor Potential of Seaweed Derived-Endophytic Fungi

**DOI:** 10.3390/antibiotics8040205

**Published:** 2019-10-31

**Authors:** Thaiz Rodrigues Teixeira, Gustavo Souza dos Santos, Lorene Armstrong, Pio Colepicolo, Hosana Maria Debonsi

**Affiliations:** 1Department of Physics and Chemistry, School of Pharmaceutical Sciences of Ribeirão Preto, University of São Paulo, RibeirãoPreto, SP 14040903, Brazil; thaiz_rt@hotmail.com (T.R.T.); gustavosouzasantoos@gmail.com (G.S.d.S.); 2Department of Pharmaceutical Sciences, State University of Ponta Grossa, Ponta Grossa, PR 84030900, Brazil; lorenearmstrong@hotmail.com; 3Department of Biochemistry, Chemistry Institute, University of São Paulo, São Paulo, SP 05508-000, Brazil; piocolep@iq.usp.br

**Keywords:** endophytic fungi, seaweed, cytotoxicity, Marine Natural Products, marine biotechnology

## Abstract

The marine environment presents a high biodiversity and a valuable source of bioactive compounds with therapeutic and biotechnological potential. Among the organisms present in marine environment, the endophytic fungi isolated from seaweed stand out. These microorganisms have aroused interest in the scientific community regarding its various activities such as antiviral, antimicrobial, antioxidant, photoprotective, cytotoxic, genotoxic, anti-inflammatory, and anticancer, besides establishing important ecological relations with its hosts. Anticancer molecules derived from marine natural sources are a promising target against different types of cancer. The disease’s high rates of morbidity and mortality affect millions of people world wild and the search for new therapeutic alternatives is needed. Thus, this review partially summarizes the methodologies for the isolation of seaweed-derived endophytic fungi, as well as describes the anticancer compounds isolated from such microorganisms, reported in the literature from 2009 to the present. In addition, it describes how some biotechnological processes can help in the discovery of bioactive compounds, especially with anticancer activity.

## 1. Introduction

Cancer cells can develop from any tissue in the human body, they multiply and grow uncontrolled. Some types of cancers can form tumors and spread throughout the body, forming metastases [1]. Talking about cancer brings a wealth of information and discussions about the disease and major challenges that still loom in the 21st century. It is a disorder with remarkable growth and high rates of morbidity and mortality. Only in 2018, it was reported about 18 million cancer cases and 10 million deaths. The most common types of cancer are lung, breast (women) and colorectal, this one with 1.8 million new cases [1,2,3]. An estimate of the International Agency for Research on Cancer (IARC), World Health Organization is that in 2020 the number will increase to approximately 1 million [4]. Despite these scary numbers, not all cancers are malignant, with reports of some patients with a survival rate of over 90% [1]. Different treatment approaches may be used for cancer, such as surgical removal, radiotherapy, immunotherapy, and chemotherapy. Chemotherapy drugs can often be ineffective, where drug resistance occurs, undesirable side effects, reducing the patient’s quality of life. Recently, new drugs have been introduced to the market to increase the treatment potential of chemotherapeutic agents [1,3]. These drugs can be derived from natural sources and, in its original structure, semi-synthetics or analogous structure [5,6,7].

Plants, microorganisms (marine and terrestrial) and marine organisms are sources of natural products with new and unique chemical structures. In addition, many of these molecules are potential anticancer agents [6,8]. The recent approaches for prospecting new compounds with bioactivity, includes extracts dereplication, in silico strategies, mathematical and statistical modeling that can predict structure-activity relationships, and provide information about the physicochemical properties of moleculesto optimize the search/isolation process for the target drug [8].

In recent years, the search for new compounds from the marine environment has constantly increased due to the discovery of distinct chemical structures and their bioactivity. There are four clinical approved drugs from natural marine products that are in use for the treatment of cancer: cytarabine (Cytosar-U^®^), it is a derivative from a marine sponge, used for leukemia; trabectidine (Yondelis^®^), isolated from a tunicate, medicinally in use for ovarian cancer and soft tissue sarcoma; eribulin mesylate (Halaven^®^, for metastatic breast cancer), a derivative from a sponge and, brentuximab vedotin (Adcetris^®^, a conjugated antibody used in Hodgkin’s lymphoma and anaplastic large T-cell malignant lymphoma), from a mollusk source. Marine-derived substances with antitumoral activities are continuously being isolated and tested. For example, lurbinectedin, plitidepsin and plinabulin (isolated from marine fungus) are currently in phases I, II, and III of clinical trials [5,7,9].

The microorganisms’ classes that will be discussed in this article are the endophytic fungi derived from seaweed. In the marine environment, they can also be hosted in sponges for example; however, there are reports that their presence is prominent in macroalgae. Since the discovery of new substances in the marine environment, new sources are being investigated all the time, isolated fungal metabolites have shown to belong to different chemical classes, for instance, polyketides, lactones, steroids, alkaloids, terpenoids, isocoumarins, and phenols. Isolated fungal metabolites have various biological activities, such as antiviral, antimicrobial, antioxidant, photoprotective, cytotoxic, genotoxic, anti-inflammatory, anticancer and kinase inhibitors (drug targets for a number of therapeutic areas) [10,11,12,13,14].

Cytotoxic metabolites from endophytic fungi are well represented for the dimeric diketopirerazines, as an example we can cite the leptosins, which are inhibitors of the DNA topoisomerases (I and II) and, great candidates for anticancer drugs. They were isolated from the fungus *Leptoshaeria* sp., an endophyte of the macroalgae *Sargassum tortile*. Terpenoids, lactones, and alkaloids isolated from marine fungi also have cytotoxic activity against leukemia, HCT-116 colon carcinoma, A549 lung cancer cells and others [13,15].

The aim of this review is to report the findings about anticancer compounds isolated from seaweed derived-endophytic fungi in the last ten years (2009-2019) which have shown bioactivity against different types of cancer cell lines: NCI-H460 and NCI-H446 (human lung carcinoma), A-549 (adenocarcinomic human alveolar basal epithelial), MDA-MB-231 (human breast cancer), HCT-116 (human colon carcinoma), etc. Herein, we report the compounds from a variety species of macroalgae including *Laurencia*, *Sargassum* and *Pterocladiella* and their endophytes, especially, *Aspergillus*, *Cladosporium* and *Penicillium*. The secondary metabolites that are shown in the current paper present in vitro and/or in vivo studies tested in different cancer cells lines. We also summarize some mechanisms of action and biotechnological processes to obtain anticancer natural products. Due to the high incidence rate of cancer, resistance to many chemotherapeutic drugs and their adverse effects on the patient, as mentioned above, the relentless search for new natural products leads becomes necessary, in order to alleviate these unpleasant symptoms that occur during treatment and increase the chances of cure of the patient.

## 2. Endophytic Fungi

Endophytic fungi are a polyphyletic group comprising mainly Ascomycetous fungi [16], which have been on the focus of research interest of mycologists, chemists and pharmacists regarding to its ecological importance and biotechnological applications. In the past decades, much work has been done regarding the study of these microorganisms and many definitions for this ecological group has been proposed.

Petrini [17] defined endophytes as all microorganisms inhabiting plant organs that, at some time of their life, can colonize internal plant tissues without causing apparent harm to the host. Wilson [18] described the term “endophyte” as any organism living within plant tissues (Gr. Endon, within; phyton, plant) without manifestation of disease symptoms [19]. For Schulz and Boyle [20] fungal endophytes are colonizers of plant tissues that do not cause any visible disease symptoms to the host at any specific moment. Despite the definitions for the term and the word’s etymology reference to these microorganisms as plant associated only, currently the term endophyte has been used as synonym of mutualistic [21]. Fungal endophytes can be found in all climatic conditions in obligate or facultative relations. The lack of precise information points the need to understand the coexisting behavior between endophytes and their hosts and why host defenses does not work to eliminate the colonizing endophytes [19]. This might be related with the latent stage endophytes assume once they enter the host organism. However, endophytes can be influenced by environmental conditions or ontogenetic state of the host to become pathogenic.

All plants in natural ecosystems studied to date appear to be symbiotic with fungal endophytes [22,23]. Regarding to the marine environment, endophytes can be found in associations with seaweed and represent an ecologically defined group of marine-derived organisms denominated Marine Algicolous Fungi (MAFs) [24]. These organisms benefit from this ecological relation usually getting nutrition and protection from the host organism, in return, endophytes play important roles in the ecological adaptation of their host, such as increasing environmental stress tolerance, improving plant vigor and decreasing herbivore attack by producing certain secondary metabolites [25,26].

### 2.1. Ecological Role of Endophytic Fungi

Fungal associations with land plants date back from early evolutionary times [27]. Symbionts can affect host ecology, capacity of adaptation, and evolution [28], which includes the ability to form host communities, community structure and diversity of associated microorganisms [23].

Macroalgae are a prolific source of natural products with industrial interest and can have the secondary metabolism influenced by associated microorganisms such as endophytic fungi and vice-versa [29]. As it occurs in land plants, marine derived endophytic fungi were found to have physiological and ecological roles for the fungal-host interaction, which comprise nutritional enhancement, stabilization of host skeleton, and secondary metabolite production [30]. For example, Zuccaro et al. [31] reported a new species of *Acremonium* spp., *Acremonium fuci* which the conidia were only capable to germinate in the presence of *Fucus serratus* tissue or its aqueous tissue homogenates, and not in seawater alone. The production of these metabolites provides chemical adaptation to environmental conditions, substrate competitionand, also it helps to protect the host against pathogens attacks [19]. A recent study also reported a possible beneficial association of endophytic fungi and a marine brown alga, where the metabolites pyrenocines were able to protect the algae *Pyropia yezoensis* against the infection by protistan pathogens of marine algae, highlighting the importance of this symbiosis [32].

Many macroalgal species have been studied worldwide regarding their associated fungal communities, which includes the genera *Ascophyllum, Ballia, Caulerpa, Ceramium, Ceratiodictyon, Cladophora, Chondrus, Dictyota, Dilsea, Egregia, Enteromorpha, Fucus, Gelidiella, Gracilaria, Grateloupia, Halimeda, Halymenia, Hypnea, Laminaria, Lobophora, Padina, Porphyra, Portieria, Saccorhiza, Sargassum, Stoechospermum, Turbinaria*, and *Ulva* [33].

This review provides information concerning nine macroalgae genera, being *Sargassum* and *Laurencia* the most studied to date, followed by *Enteromorpha, Ulva, Codium, Grateloupia, Leathesia, Pterocladiella,* and *Undaria.* Associated to these genera, we can correlate 10 genera of endophytic fungi, which includes *Aspergillus, Cladosporium, Paecilomyces, Chaetomium, Penicillium, Guignardia, Phoma, Talaromyces, Gibberella, Coniothyrium*, being *Aspergillus* and *Penicillium* the prevalent ones (Table 1). Despite the number of studies addressing fungal endophytes from macroalgae, these works focus mainly in the secondary metabolite production of these organisms, which highlights the need for studies regarding the ecological roles of these microorganisms in macroalgae hosts.

### 2.2. Isolation of Endophytic Fungi

Since fungal endophytes live within host tissues, the isolation of these microorganisms is a method-dependent process [34]. The sterilization method has much influence in the microorganism’s isolation and, the techniques are diverse for each host organism being studied. Distinct hosts require different sterilization times, for example, thicker leaves require longer times and extreme sterilization conditions than thin leaves [22]. Epiphytic contaminants may interfere in the study if the sterilization method is not efficient, on the other hand the method cannot be very aggressive, and otherwise the number of isolates will be reduced, since it caused damage to the host tissues at the beginning of the process.

A control is also needed to make sure the fungi growing on the fragments are truly endophytes. Schulz et al. [35] developed a control method for discard epiphytic fungi. The method consists on making leaves imprints on the agar plate surface. The absence of fungi growing out on the imprinted agar plate indicates that the sterilization method can be considered effective. This review combined 24 studies regarding the marine fungi endophytes isolation, among them, we found out three variations of sterilization methodologies (Table 1), differing in time and sterility application.

Our research group developed a protocol for fungi endophytes isolation from tropical and polar macroalgae, from adaptations of the study of methodologies published by Erbert and co-workers [36], based on the publication of Kjer and co-workers [37]. The protocol consists in the utilization of three different sterilization methods, which begins with the macroalgae rinsing in distilled seawater followed by: (i) 15 seconds in ethanol (70% *v/v*), washing off macroalgae fragments in three different distilled water recipients to remove sterilizers residue; (ii) 5 seconds in ethanol 70% (*v/v*) followed by 5 seconds in sodium hypochlorite (2.5% *v/v*); and (iii) 5 seconds in ethanol 70% (*v/v*) followed by 10 seconds in sodium hypochlorite (2.5% *v/v*), after the sterilization procedure macroalgae segments are used to make imprints on agar surface (negative control 1). Macroalgae is then aseptically cut into fragments (0.5 mm to 1 cm) and placed in agar plates. The third distilled water of the washing procedure is also inoculated in agar plates, aiming to check no microorganism’s growth (negative control 2). To the selective isolation of fungi, the culture medium utilized is supplemented with chloramphenicol (200 mg·L^−1^) [38].

## 3. Cytotoxic Secondary Metabolites Produced by Endophytic Fungi

Fungi as well as bacteria, are versatile organisms that can be found in diverse habitats, occupying even inhospitable ecological niches, of all ecosystems around the planet. The ecological role of these organisms goes far beyond interactions between living beings, as they can adapt to virtually any environment and promote new biological interactions from chemical communications. The association between fungi and marine algae promotes the biosynthesis of metabolites with therapeutic potential by these microorganisms [62,63,64]. Currently, a large number of unique chemical structures with biological and pharmacological activities have been isolated of fungi from the marine environment and despite the absence of metabolites derived from theses microorganisms in the clinical pipeline, dozens of them have been classified as potential chemotherapy candidates [65]. Among these compounds, the chemical classes that have been seen are: polyketides, alkaloids, peptides, lactones, terpenes, and sterols. Distinct substances can be mentioned: asperpyrone A-D [49], cladosporols F−I [51], phomaketides A-E [60], cytoglobosins C-D [50], variloid A-B [56], cyclo-(Tyr-Leu), cyclo-(Phe-Pro) [54], insulicolide A [40], asperolide A-C, wentilactone A-B [44], penicisteroids A-B [57,58] and several others that were isolated from different species of fungi and can beseen on Table 1. The antitumor activity of these compounds was evaluated against several tumor cell lines, such as HeLa (human epithelial carcinoma), A-549 (adenocarcinomic human alveolar basal epithelial), HepG2 (human hepatocellular carcinoma), NCI-H460 and NCI-H446 (human lung carcinoma), SMMC-7721 (human hepatocarcinoma), SW1990 (human pancreatic cancer), MCF-7 (human breast adenocarcinoma), MDA-MB-231 (human breast cancer, HCT-116 (human colon carcinoma), PANC-1 (human pancreatic cancer), Caco-2 (human colorectal adenocarcinoma), Huh-7 (human hepatocarcinoma), DU145 (human prostate cancer), HL-60 (human leukemia), and others (Table 1).

The chemical and biological potential of fungi of marine origin in the search for new structures with promising antitumor activities led to the identification of several relevant compounds. Over the past few years, several metabolites produced by seaweed derived-endophytic fungi or other organisms have shown potent antitumor effects, which have been evaluated by different mechanisms, such as the ability to kill cancer cells with low or no toxicity to healthy host cells, blocking key enzymes, stimulating the pathways of death, or promoting growth arrest [65,66]. The current review, which covers the period 2009 to 2019, emphasizes the isolated metabolites from seaweed derived-endophytic fungi, that were able to inhibit growth different cancer cells.

### 3.1. Alkaloids and Nitrogen-containing Heterocycles

Cytochalasans are fungal alkaloids with biological activities cytoskeletal processes, including cytotoxicity against tumor cell lines. Cytoglobosins C and D **(1-2)** (Figure 1), cytochalasan derivatives, isolated from the cultures of *Chaetominum globosum* QEN-14, an endophytic fungus derived from the marine green algae *Ulva pertusa* showed moderate cytotoxic activity against adenocarcinomic human alveolar basal epithelial cell A-549 (IC_50_ 2.26 and 2.55 µM, respectively) [50].

Polyketides-type alkaloids (−)−cereolactam **(3)** and (−)−cereoaldomine **(4)** (Figure 1) are phenalenone derivates that selectively inhibit the human leukocyte elastase (HLE) with IC_50_ values of 9.28 and 3.01 µM, respectively. These metabolites are as unprecedented structural types and may be formed by the biosynthetic degradation of phenalenone-type precursors. Compounds **3** and **4** were isolated from the *Coniothyrium cereale*, an endophytic fungus derived from marine green algae *Enteromorpha* sp. [67].

A pyrrolidine derivative,3-hydroxy-5-(hydroxymethyl)-4-(4’-hydroxyphenoxy)pyrrolidin-2-one **(5)** (Figure 1) was isolated from the cultures of *Gibberella zeae*, an endophytic fungus isolated from the marine green alga *Codium fragile*. Compound **5** showed that it possessed 61.80% and 17.60% inhibitory rates against A-549 and BEL-7402 tumor cell lines at 10 µM, respectively [68].

Prenylated indole alkaloids are a class of secondary metabolites commonly found in filamentous fungi, especially in the genus *Penicillium* and *Aspergillus*. Some of these compounds have insecticidal, cytotoxic, anthelmintic and antibacterial activities. Two prenylated indole alkaloids, dihydrocarneamide A **(6)** and iso-notoamide B **(7)** (Figure 1) were isolated from marine-derived endophytic fungus *Paecilomyces variotii* EN-291. These compounds were assayed for their cytotoxic activities against human large cell lung carcinoma cell line (NCI-H460) and showed weak activity with IC_50_ values of 69.30 and 55.90 µmol L^−1^, respectively [69]. The indole alkaloids, varioloid A **(8)** and B **(9)** (Figure 1) were also isolated from the marine alga-derived endophytic fungus *Paecilomyces variotii* EN-291. Compound **8** showed potent cytotoxicity against A-549, HCT116, and HepG2 cell lines, with IC_50_ values of 3.50, 6.40, and 2.50 μgmL^−1^, respectively, while compound **9** also showed considerable activities, with IC_50_ values of 4.60, 8.20, and 6.60 μg mL^−1^, respectively [56].

From the culture of the endophytic fungus *Guignardia* sp. isolated from brown algae *Undaria pinnatifida* (Harv.) Sur. collected in Changdao sea area, China, two of the five peptides that were isolated, cyclo-(Tyr-Leu) **(10)**, cyclo-(Phe-Pro) **(11)** (Figure 1) presented cytotoxic activity. Both compounds exhibited activity against KB cell line with IC_50_ of 10.00 μg mL^−1^, comparable to that of 5-fluorouracil (2.50 μg mL^−1^) co-assayed as a positive reference [54].

### 3.2. Polyketides

As an example of polyketides with anticancer potential, natural naphthopyrones isolated from the endophytic fungus *Aspergillus* sp. XNM-4 derived from brown algae *Leathesia nana* are highlighted. Compounds asperpyrone B **(12)**, aurasperone F **(13)** and especially asperpyrone A **(14)** (Figure 1) exhibited potent cytotoxicity on PANC-1, A-549, MDA-MB-231, Caco-2, SK-OV-3 and Hl-7702 cells. Compound **(14)** possessed the greatest inhibitory effects against PANC-1, with an IC_50_ value of 8.25 ± 2.20 µM [49].

Four new cladosporol derivatives, clodosporols F-I, the know clodosporol C and its new epimer, cladosporol J were isolated and identified from the marine algal-derived endophytic fungus *Cladosporium cladosporioides* EN-399. All compounds were assayed for cytotoxicity activity. Cladodosporol H **(15)** (Figure 1) exhibited significant cytotoxicity against A-549, Huh7 and LM3 cell lines with IC_50_ values of 5.00, 1.00, and 4.10 µM, respectively, clodosporol C **(16)** (Figure 1) showed activity against H446 cell line with IC_50_ value of 4.00 µM [51]. Cladosporols are known to be unique to the fungal genus *Cladosporium* and have attracted the attention of researchers because of their ability to stimulate cell cycle G1-phase arrest in human colon carcinoma HT-29 cells, thus demonstrating their promising antitumor activity [51,70,71,72].

From the culture of the marine-derived fungus *Coniothyrium cereale* isolated from the green alga *Enteromorpha* sp. collected from Fehmarn, Baltic Sea, seven new phenalenone derivatives as well as known compounds were isolated and assayed for cytotoxicity activity against human urinary bladder carcinoma cells 5637 (HTB-9) and HLE. The isolated compounds, coniosclerodin **(17)**, conioscleroderolide **(18)** and coniolactone **(19)** (Figure 1) showed potent inhibition against HLE cell line with IC_50_ values of 7.20, 13.30 and 10.90 µM, respectively [52].

Penicitide A **(20)** (Figure 1) was isolated from the *Penicillium chrysogenum* QEN-24S, an endophytic fungus isolated from an unidentified marine red algal species of the genus *Laurencia*. Compound **20** exhibited moderate cytotoxic activity against the human hepatocellular liver carcinoma (HepG2) cell line with IC_50_ value of 32.00 µg mL^−1^ [57].

Seven new polyketides, phomaketides A-E, pseurotins A_3_ and G, besides 11 known compounds were purified from endophytic fungal strain *Phoma* sp. NTOU4195 isolated from the marine red alga *Pterocladiella capillacea* and evaluated for their antiangiogenic activity. Angiogenesis is defined as the formation of blood vessels from an existing vascular network towards a tumor, is crucial for the progression of the disease in tumor types. Thus, the antitumor activity of compounds isolated from *Phoma* sp. was assessed by the ability to prevent or not the capillary-like tube formation, which is one of the most important steps in angiogenesis. Compound phomaketide A **(21)** (Figure 1) exhibited the most potent antiangiogenic activity by suppressing the tube formation of endothelial progenitor cells (EPCs) with an IC_50_ of 8.10 µM. The others compound phomaketide C **(22)**, D **(23)**, E **(24)** and pseurotin G **(25)** (Figure 1) showed weak activity with IC_50_ of 17.80, 16.20, 19.20 and 16.70 µM, respectively [60].

### 3.3. Quinones

A new metabolite named isorhodoptilometrin-1-methy ether **(26)** (Figure 1) along with the known compounds emodin, 1-methyl emodin, evariquinone, 7-hydroxyemodin 6,8-methyl ether, siderin, arugosin C and variculanol were isolated from endophytic fungal strain *Aspergillus versicolor*, reported as endophyte from the inner tissue of the green alga *Halimeda opuntia*. Compound **26** demonstrated mild solid tumor selectivity HepG2 compared to the human normal cells (CFU-GM) when 3 µg of the pure compound was applied to the filter disk, while compounds emodin **(27)** and variculanol **(28)** (Figure 1) showed weak activity against HCT-116 and HepG2, respectively [43].

Five new polyhydroxylated hydroanthraquinone derivatives were isolated and identified from the culture extract of *Talaromycesis landicus* EN-501, an endophytic fungus obtained from the inner tissue of the marine red alga *Laurencia okamurai*. These compounds were assayed for cytotoxicity against sensitive and cisplatin-resistant human ovarian cancer cell lines A2780 and A2780 CisR, respectively. However, none of the compounds were active (IC_50_ < 10 µM) [61].

### 3.4. Terpenoids and Sterols

#### 3.4.1. Terpenoids

The chemical investigation of the endophytic fungus *Aspergillus ochraceus* Jcma1F17, which was isolated from a marine alga *Coelarthrum* sp. resulted in the isolation of a new nitrobenzoyl sesquiterpenoid, 6*β*, 9*α*-dihydroxy-14-*p*-nitrobenzoylcinnamolide **(29)**, and a known analogue, insulicolide A **(30)** (Figure 1). The compounds were evaluated for their cytotoxic activity against H1975, U937, K562, BGC-823, Molt-4, MCF-7, A-549, Hela, HL60 and Huh-7. Compounds **29** and **30** displayed significant cytotoxicity against 10 human cancer cell lines, with IC_50_ values of 1.95 to 6.35 µM [40]. There are few nitrobenzoyl sesquiterpenoids reported in the literature, most of them being obtained from fungi of the genus *Aspergillus* of marine origin that were isolated from sediment [73], and from marine organisms such as seaweed [40,41,72] and sponges [73].

Two new nitrobenzoyl sesquiterpenoids, insulicolide B and C were also isolated from culture extracts of the marine-derived fungus *A. ochraceus* Jcma1F17, together with three known nitrobenzoyl sesquiterpenoids and a derivative sesquiterpenoid. All compounds were evaluated for their cytotoxicity against three renal carcinoma cell lines, ACHN, OS-RC-2, and 786-O cells. Compounds 14-*O*-acetylinsulicolide A **(31)**, 6*β*,9*α*-dihydroxy-14-*p*-nitrobenzoylcinnamolide **(29)** and insulicolide A **(30)** (Figure 1) displayed activities with IC_50_ values of 0.89 to 8.20 µM. Further studies indicated that compound **31** arrested the cell cycle at the G0/G1 phase at a concentration of 1 μM and induced late apoptosis at a concentration of 2.00 μM after a 72-h treatment of 786-O cells [41]. Therefore, the nitrobenzoyl sesquiterpenoids have attracted attention of researchers, because of their good antitumor potential.

Three new tetranorlabdane diterpenoids, asperolides A–C and five related derivatives (a tetranorditerpenoid derivative, wentilactones A and B, botryosphaerin B and LL-Z1271-β) were obtained of the culture extract of *Aspergillus wentii* EN-48, an endophytic fungus isolated from an unidentified marine brown alga species of the genus *Sargassum*. All compounds were assayed for their cytotoxic activities against HeLa, HepG2, MCF-7, MDA-MB-231, NCI-H460, SMMC-7721 and SW1990 tumor cell lines, with fluorouracil and adriamycin as positive controls. None of these compounds had significant activity (IC_50_ ≤ 10 µM). Wentilactone B **(32)** (Figure 1) was the most potent among the tested compounds with IC_50_ value of 17.00 µM against SMMC-7721 cell line [44].

Other researchers have explored the promising antitumor activity of compound **32**. Zhang et al. [46] demonstrated that wentilactone B **(32)** could efficiently induce SMMC-7721 cells apoptosis, but not normal hepatic cells and inhibit the metastasis of this cell line. In addition, further studies have shown that wentilactone B **(32)** can significantly induce cell cycle arrest at G2 phase and mitochondrial-related apoptoses, accompanying the accumulation of reactive oxygen species (ROS). Therefore, this agent may be a potentially useful compound for developing anticancer agents for hepatocellular carcinoma [45].

Further studies carried out with wentilactone A **(33)**, isolated from endophytic fungus *A. wentii* EN-48 [44], demonstrated that compound **33** triggered G2/M phase arrest and mitochondrial-related apoptosis in human lung carcinoma cells (NCI-H460 and NCI-H446), accompanying the ROS accumulation. In vivo studies, wentilactone A **(33)** (Figure 1) suppresses tumor growth without adverse toxicity and presented the same mechanism as that in vitro [47]. Asperolide A **(34)** (Figure 1) isolated from the endophytic fungus *A. wentii* EN-48 as previously mentioned [44] was also evaluated for its antitumor activity in NCI-H460 cells. Compound **34** leads to the inhibition of NCI-H460 lung carcinoma cell proliferation by G2/M arrest with the activation of the Ras/Raf/MEK/ERK signaling and p53-dependent p21 pathway. An in vivo study with asperolide A **(34)** illustrated a marked inhibition of tumor growth, and little toxicity compared to cisplatin therapy, which proves its potential antitumor activity [48].

#### 3.4.2. Sterols

Two new steroid derivatives, 3*β*, 11*α*-dihydroxyergosta-8,24(28)-dien-7-one **(35)** and 3*β*-hydroxyergosta-8,24(28)-dien-7-one, and a rare 7-norsteroid with an un usual pentalactone B-ring system, the 7-Nor-esgosterolide **(36)** (Figure 1) were characterized from the culture extract of *Aspergillus ochraceus* EN-31, an endophytic fungus isolated from the marine brown alga *Sargassum kjellmanianum*. Others nine known related steroids were isolated and identified. The steroids cited were assayed for their cytotoxic activities against NCI-H460, SMMC-7721, SW1990, DU145, HepG2, HeLa and MCF-7 tumor cell lines. Compound **35** displayed selective cytotoxic activity against the SMMC-7721 cell line with an IC_50_ value of 28.00 μg mL^−1^, while compound **36** exhibited selective cytotoxic activity against NCI-H460, SMMC-7721 and SW1990 cell lines with IC_50_ values of 5.00, 7.00 and 28.00 μg mL^−1^, respectively [39].

From the culture of the endophytic fungus *Guignardia* sp. isolated from brown algae *Undaria pinnatifida* (Harv.) Sur. collected in Changdao sea area, China, ergosterol peroxide, 6,22-diene-5,8-epidioxyergosta-3-ol **(37)** (Figure 1) and ergosterol were isolated and assayed for cytotoxic activity against KB cell line. Compound **37** exhibited activity with IC_50_ of 20.00 μg mL^−1^ [54].

Penicisteroids A **(38)** and B **(39)**, two new polyoxygenated steroids, together with seven known steroids were obtained from the culture extract of *Penicillium chrysogenum* QEN-24S, an endophytic fungus isolated from unidentified marine red algal species of the genus *Laurencia*. The cytotoxicity against seven tumor cells was determined and compound **38** displayed selective activity against the tumor cells line HeLa, SW1990 and NCI-H460 with the IC_50_ of 15.00, 31.00 and 40.00 μg mL^−1^, respectively, while the other compounds displayed weak or no appreciable activity. The hydroxyl group at C-6 in B-ring seems essential for their cytotoxicity, which is likelythe reason for that penicisteroid A **(38)** (Figure 1) showed cytotoxic activityagainst the cell lines HeLa, SW1990, and NCI-H460, while penicisteroid B **(39)** (Figure 1) showed no activity [58].

### 3.5. Others 

From the culture of the fungus *Aspergillus tennesseensis*, a marine algal-derived endophytic fungus isolated from the fresh tissue of an unidentified marine alga, were isolated two new compounds with a prenylated diphenyl ether structure, diorcinol *L* and (*R*)-diorcinol B, along seven known compounds. All compounds were evaluated for cytotoxicity against six tumor cells lines (A-549, Du145, HeLa, MCF-7, MDA-MB-231 and THP-1) in vitro. Compound 3-(2-(1-hydroxy-1-methyl-ethyl)-6-methyl-2,3-dihydrobenzofuran-4-yloxy)-5-methylphenol **(40)** (Figure 1) selectively exhibited cytotoxicity against the THP-1 cell line with the IC_50_ value of 7.00 μg mL^−1^, while others displayed weak or no inhibitory activity (IC_50_> 50 μg mL^−1^) [42].

A new chromone derivatives, 2-(hydroxymethyl)-8-methoxy-3-methyl-4H-chromen-4-one (chromanone A) **(41)** (Figure 1), was obtained from de *Penicillium* sp., an endophytic fungus isolated from seaweed *Ulva* sp. The researchers tested the modulatory effect of compound **41** on carcinogen metabolizing enzymes. Carcinogens is activated by cytochrome P-450 1A (CYP1A) and detoxified by glutathione S-transferases (GST), quinine reductase (QR) and epoxide hydrolase (mEH). The researchers demonstrated in their studies that chromanone A is a promising inhibitor of CYP1A activity up to 60% of the stimulated-CYP1A in Hepa1c1c7 cells, and it significantly induced GST but not total thiols at low concentrations. In addition, chromanone A had influence on QR activity, while it resulted in a significant dose-dependent echancement mEH activity in Hepa1c1c7 cells. Therefore, chromanone A may act as an active tumoranti-initiating via modulation of carcinogen metabolizing enzymes and protection from DNA damage [59].

## 4. Biotechnology of Marine Endophytic Fungi

As previously reported endophytic fungi of marine origin have a promising therapeutic and biotechnological potential. Bioprospecting and the development of these products require sustainable processes that substantially increase the biomass of such microorganisms. Thus, marine biotechnology can contribute significantly to the production of these bioactive metabolites at various levels of the process, including obtaining, production, processing and development. However, biotechnological protocols applied to obtain bioactive metabolites of marine fungi are still quite deficient [74].

The methods and technologies applied in marine fungal biotechnology derive largely from terrestrial fungi processes and rarely reflect the specific demands of fungi of marine origin [74]. The most famous example of biotechnology applied to endophytic fungi is taxol, a multibillion-dollar anticancer compound produced in yew plant *Taxus brevifolia* by the terrestrial endophytic fungus *Taxomyces andrenae* [75,76]. The fascinating discoveries from studies with terrestrial endophytic fungi motive the studies of theses microorganisms of marine origin, mainly those isolated from seaweed. However, the ability to produce metabolites of therapeutic and/or biotechnological interest by endophytic fungi from marine algae has been underestimated, since many genes related to the biosynthesis of these substances are silenced in artificial laboratory culture conditions. Some strategies were applied to activate these silent gene clusters in filamentous endophytic fungi, for instance, the co-culture of microorganisms, mimetizing the original ecosystem is a possibility to stilted the production of natural products. Besides that, the epigenetic manipulation as well as allow the possibility to obtain new compounds and, through gene inactivation (“knock outs”), transcription factors, activation by the over expression and deletion of genes [77,78].

### Biotechnological Processes to Obtain Bioactive Metabolites of Endophytic Fungi

In general, for the discovery of a new product with therapeutic, alimentary, cosmetic or biotechnological potential, a species of endophytic fungus is selected, its metabolites are extracted, it is carried out the screening for several activities (bioactivity-guided), process of isolation and the natural product is obtained in its pure form. This strategy of discovery is slow, tedious, intensely laborious and sometimes inefficient [79]. Dereplication and analytical methods are great strategies that are used to preview and to explore new secondary metabolites. Advances in this area of research have provided new strategies this discovery. It is possible to recognize by platforms (SMURF, antiSMASH, Fun-Gene Clusters) the gene clusters, which are responsible to synthesize individual groups of compounds, a disadvantage of this technique, is the inactivation of genes in the laboratory conditions as mentioned above. Starting with crude extracts and fractions it is possible to develop a sequence to dereplicate natural products. After the extraction and fractionation, the samples are analyzed indifferent equipments: LC-MS, and LC-MS/MS, UV, IR and NMR spectra, etc. The results are compared with known compounds present in databases (MarinLit, MassBank, MetLin, AntiBase, DNP, DMNP, PubChem) [80,81]. Kildgaard et al. [82] used the dereplication technique to analyze compounds from bioactive marine-derived fungi. The authors showed an integrated approach using ultra-high-performance liquid chromatography-diode array detection-quadrupole time off light mass spectrometry (UHPLC-DAD-QTOFMS) and identified polyketides, non-ribosomal peptides, terpenes, meroterpenoids and four novel isomers of the known anticancer compound asperphenamate, from marine-derived strains of *Aspergillus*, *Penicillium* and *Emericellopsis*. Another example of using the dereplication technique can be observed in the work of El-Elimat et al. [83], who analyzed fungal secondary metabolites in culture extracts using LC/DAD/MS with MS/MS spectral database for to search novel potent anticancer compounds before engaging in isolation process. An excellent tool for known and unknown compounds is the GNPS (Global Natural Products Social molecular networking), where MS/MS spectra are uploaded and, through the MS/MS are grouped forming chemically similar clusters. Thereafter, the known compounds are recognized by GNPS and nodes of unknown compounds can be selected as the target of possible new metabolites [80,81,84,85,86].

Biotechnology can help to accelerate the discovery of natural products of endophytic fungi of marine origin [74]. Silber et al. [74] describe some strategies for the application of marine biotechnology to the discovery of new compounds with antimicrobial activity and a large part of these strategies can be applied to other therapeutic activities or biotechnological processes. For example: (i) controlled miniaturization to increase screening efficiency, that is, the small-scale culture in fermentation systems using microtiter plates [87,88,89] or specialized miniaturized fermentation systems (System Duetz or BioLector) [90,91,92]; (ii) directed stimulus of strains to expand the chemodiversity, because the proper understanding of the endophytic fungi that produce the bioactive substances and their ecological role help to find the ideal conditions of cultivation and production and can extend the chemical diversity [74,93]; (iii) metabolomic, proteomic, and transcriptomic studies in order to elucidate the metabolic state of the cells and indicating regulatory sites at the levels of DNA, RNA and protein; from this knowledge it is possible to induce and direct the biosynthetic process and later to control it [94,95]. A comparative proteomic study of a fungus of marine origin *Microascus brevicaulis* revealed how the biotechnological fermentation process should be controlled in order to increase the production of anticancer compounds, scopularides A and B [96].

One of the major challenges for those working with natural products, regardless of source is obtaining sufficient amounts of the metabolite of interest for drug development. Biotechnological approaches such as the production using full fermentative process can assist in obtaining these products. That is, large-scale cultivation (bioreactors) using different approaches to optimize culture media and induce increased production of target molecules are an alternative to this problem [79]. However, the research addressing the optimization of marine fungal fermentation in bioreactors is insufficient [97]. The use of bioreactors allows the control of crucial factors such as aeration, dissolved oxygen, carbon dioxide, pH, salinity, temperature, foaming and growth under agitation or in static mode, among others. The control and understanding of such parameters is the basis for successful strategies on a larger scale [98]. Xu et al. [99] optimized the static culture conditions of *Aspergillus wentii* EM-48, a fungus isolated from brown seaweed in order to increase the production of asperid A. Factors such as salinity, initial pH, temperature, culture time and addition of plant growth regulators were evaluated in the study cited. Zhou et al. [100] increased biosynthesis of compound 1403C (also called SZ-6858) synthesized by mangrove endophytic fungus *Halorosellinia* sp. in bioreactor fermentation. Medium, culture pH, agitation speed, impeller type, inoculum level and dissolved oxygen were considered and systematically optimized, and an integrated nutrition and bioprocess strategy was established. The resulting 1403C production reached 2.07 g·L^−1^, which was 143.5% higher than the original production. Another study showed that the distribution of anticancer 1403C in the fermentation broth of the fungus *Halorosellinia* sp. was closely related to pH variations. The 1403C levels in the supernatant decreased while in mycelium increased with increasing pH. Thus, pH regulation was proposed and applied in order to accumulate the compound in the mycelium prior to broth extraction [101].

In addition to the physical-chemical parameters that must be controlled, the morphology also influences the biosynthesis of the target product in filamentous fungi of marine origin. Filamentous fungi have the ability to grow in different morphological appearances, ranging from dispersed filaments to highly dense networks of mycelia (pellets). Therefore, depending on the morphology presented by the fungus under study, effects such as oxygenation and nutrient levels may alter the biosynthesis of these microorganisms. For example, it was observed that in higher speeds for the fungus *Penicillium chysogenum* showed some consequences, like as, a decreasing in penicillin productions (1250 and 1500 rpm) and elevation in cell growth (100–900 rpm), besides restriction of oxygen in 700 rpm, however, this does not mean that it diminishes the production of other substances for the fungus *P. chysogenum* [102]. These and other parameters should be studied and explored in transferring crops to larger scales as in bioreactors.

When the use of bioreactors cannot be carried out in an economically feasible manner and there is a structural chemical complexity limiting the synthesis of the product of interest, semi-synthetic processes offer an alternative pathway in the development of natural products derived from marine fungi [74]. In the semi-synthesis, precursor molecules obtained by fermentation process are processed by synthesis, or a synthetic product is modified by bioconversion using enzymes, whole cells or even fermentative processes [103]. Jesus et al. [104] report the bio-oxidation of *rac*-camphor using whole cells of marine-derived fungus *Botryosphaeria* sp. isolated from marine alga *Bostrychia radicans* obtaining mainly (49%) 6-*endo*-hydroxycamphor and other products. These products are important for use in pharmaceutical formulations and these chemical modifications usually cannot be done by conventional organic synthesis.

In addition to the biotechnological applications already described (bioreactors and semi-synthesis) for the production of bioactive compounds from marine fungi, there isalso the use of heterologous systems and genetic and metabolic engineering [74]. For example, transfer of DNA from the environment to a host strain may allow the production of novel compounds [105]. The use of molecular techniques provides an alternative for heterologous expression of "silent" gene aggregates and directed manipulation of biosynthetic genes, optimizing the production process of the target compound. However, these techniques are still limited because of the complexity and size of the genetic groups encoding natural products. An example is the reprogramming of the biosynthetic pathway leading to the over expression of ennantianins, isolated from *Halosarpheia* sp. [74]. A review considering silent biosynthetic gene clusters activation brings several compounds isolated from fungi, as terraquinone A from *Aspergillus terreus*, emodin from *A. nidulans*, new cladochrome congener from *Cladosporium cladosporides* and nygerone A from *A. niger* [106]. Another study performed with *Glarea lozoyensis*, led to the isolation of pneumocandin derivatives, such as caspofungin from the pneumocandin B0 pathway. The new substances obtained show more pharmacological properties which are important for drug discovery [107]. In relation to genetic and metabolic engineering, the improvement of high yield lineages and the cloning of genes involved in the biosynthetic pathway allow, for example, to increase the flow of primary metabolites in the production of secondary metabolites. The greatest success was the optimization of the fermentation process and breeding of *Acremonium chrysogenum* strains in the production of cephalosporin C [108].

After the production process of the metabolites, for example in bioreactors, the next step in the development of a product of therapeutic interest by biotechnological procedures is “downstream processing” (DSP). This process comprises the steps of separating, disrupting, capturing and concentrating the cells, as well as extracting, purifying, refining, and obtaining the final product. Few authors report these DSP processes in fungal biotechnology, a classic example is the enzymatic treatment as part of the purification of cephalosporin [74,108]. The most effective process with cost, energy and space reduction is one that applies continuous “upstream and downstream processing”, but this procedure seems to be empirical and there are still no standards that are suitable for all chemical classes to be obtained [109,110]. From the purified final product, other processes related to medicinal chemistry can be performed, such as structure-activity relationship (SAR) studies and mechanism of action. These works help to improve bioavailability and to understand the therapeutic application of the compound [74].

All procedures cited here to advancethe performance in the discovery, production, and procurement of a natural product from marine endophytic fungi need to be improved taking into account the ideal developmental conditions and particular characteristics of these microorganisms. Therefore, further studies related to fungal biotechnology should be carried out in order to obtain more information and improve the procedures for obtaining a quality final product.

## 5. Conclusions

Marine fungi can grow on a wide variety of substrates such as wood, sediment, sand, mangroves, corals, shellfish, and marine invertebrates, on the surface and interior of algae. Among the fungi that live associated with macroalgae, endophytic fungi have a diversity of species and stand out as an extraordinary source of new bioactive compounds, as well as compounds known from other sources. Microorganisms have advantages such as the short growth cycle, easy preservation and control of cultivation conditions, besides perform strain through mutation breeding, genetic engineering, and other means. Therefore, it can significantly increase productivity and achieve industrial production. Consequently, studying seaweed-derived endophytic fungi to produce secondary metabolites has a very broad prospect on medical research and development.

The incidence rate of cancer has been growing, due to the influence of the living environment and living habits and the increasing aging of the population. High costs, serious side-effects, and multidrug-resistance tumors accelerate the complexity and difficulty of cancer treatment. Thus, the increasing demand for anticancer drugs in the international market, the search for novel and effective molecules from endophytic fungi has become urgent. This review summarizes a total of 41 (Figure 1) compounds, which belongs to alkaloids, terpenes, steroids, polyketides, and quinones. All of them showing cytotoxic activities, produced by seaweed derived-endophytic fungi described from 2009 to 2019. In addition, the isolation techniques of the microorganisms discussed here must be well defined, as well as the procedures to obtain these metabolites. In this review, it was possible to note that few studies have detailed the forms of isolation of endophytic fungi, as well as the scarce number of papers describing the fungal biotechnology application to obtaining bioactive products from fungi of marine origin, mainly anticancer compounds.

## Figures and Tables

**Figure 1 antibiotics-08-00205-f001:**
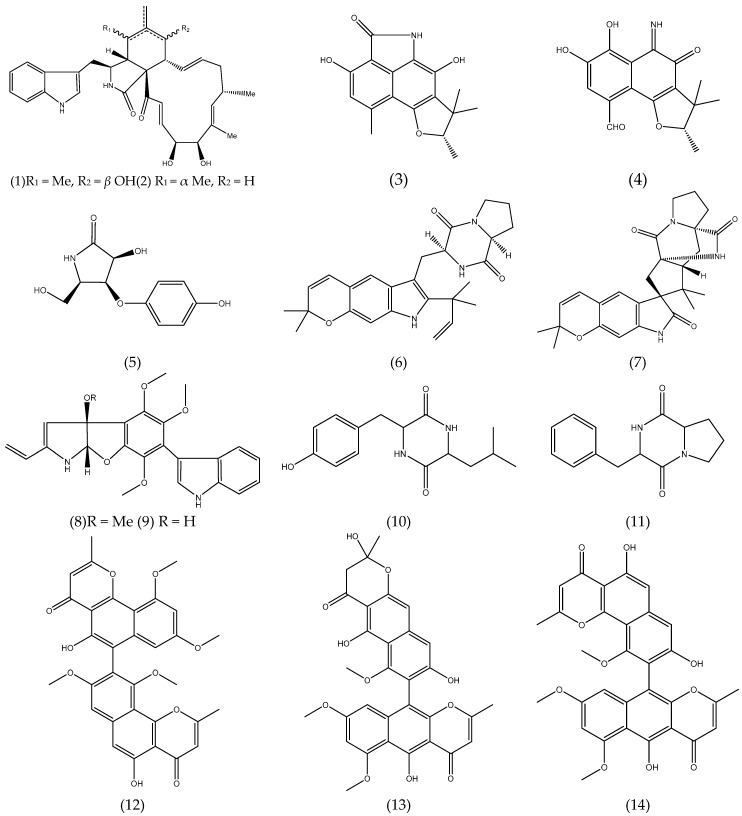
Chemical structures of natural products isolated from seaweed derived-endophytic fungi with antitumor potential published from 2009 to 2019.

**Table 1 antibiotics-08-00205-t001:** Antitumor compounds isolated from seaweed derived-endophytic fungi literature from 2009 to the present. (B) = Brown macroalgae, (G) = Green macroalgae, (R) = Red macroalgae and Rf. = Reference.

Endophytic Fungi	Sterilization Methods	Host Macroalgae	Compounds	Chemical Class	Cell line/Target Enzyme	Activity (IC_50_)	Rf
*Aspergillus ochraceus* EM-31	-	*Sargassu kjellmanianum* (B)	7-*nor*-ergosterolide; 3*β*,11α-dihydroxyergosta-8,24(28)-dien-7-one; 3*β*-hydroxyergosta-8,24(28)-dien-7-one; (22*E*,24*R*)-3*β*,5α,9α-trihydroxyergosta-7,22- dien-6-one; (22*E*,24*R*)-3*β*,5α- dihydroxyergosta-7,22-dien-6-one;ergosterol; (22*E*,24*R*)-ergosta-4,6,8(14),22-tetraen-3-one; (22*E*,24*R*)-ergosta-7,22-diene- 3*β*,5*α*,6*α*-triol; (22*E*,24*R*)-ergosta-7,22-diene-6*β*- methoxy-3*β*,5*α*-diol; (22*E*,24*R*)-ergosta-7,22- diene-3*β*,6*β*-diol; (22*E*,24*R*)-ergosta-5*α*,6*α*- epoxide-8,22-diene-3*β*,7*α*-diol; (22*E*,24*R*)-5*α*,8*α* -epidioxyergosta-6,22-dien-3*β*-ol	steroids	NCI-H460, SW1990, SMMC-7721, HeLa, DU145, HepG2, MCF-7	5.00–28.00 µg mL^−1^	[39]
*Aspergillus ochraceus* Jcma1F17	(1) Rinsed 3x with sterile sea water; (2) 60–120 s 70% EtOH; (3) Rinsed with sterile artificial sea water.	*Coelarthrum* sp. (R)	6*β*,9*α*-dihydroxy-14-*p*-nitrobenzoylcinnamolide; insulicolide A	terpenoids	H1975, U937, K562, BGC-823, Molt-4, MCF-7, A549, HeLa, HL60, Huh-7	1.95–9.40μM	[40]
*Aspergillus ochraceus* Jcma1F17	(1) Rinsed 3x with sterile sea water; (2) 60–120 s 70% EtOH; (3) Rinsed with sterile artificial sea water.	*Coelarthrum* sp. (R)	insulicolide B; 14-*O*-acetylinsulicolide A; insulicolide C;6*β*,9*α*-dihydroxy-14-*p*- nitrobenzoylcinnamolide; insulicolide A; 9-deoxyinsulicolide A	terpenoids	ACHN, OS-RC-2, 786-O	0.89–8.20 μM	[41]
*Aspergillus tennesseensis*	(1) 15 s 70% EtOH; (2) Rinsed in sterile water	Not identified	diorcinol L; (*R*)-diorcinol B; (*S*)-diorcinol B; 9-acetyldiorcinol B; diorcinol C; diorcinol D; diorcinol E; diorcinolJ; 3-(2-(1-hydroxy-1- methyl-ethyl)-6-methyl-2,3- dihydrobenzofuran-4-yloxy)-5-methylphenol	ethers	THP-1, A559, Du145 HeLa, MCF-7 MDA-MB-231	7.00–50.00 µg mL^−1^	[42]
*Aspergillus versicolor*	-	*Halimeda opuntia* (G)	isorhodoptilometrin-1-methyl ether; emodin; 1-methyl emodin;evariquinone; 7-hydroxyemodin-6,8-methyl ether; siderin; arugosin C; variculanol	quinines	Murine L1210, CCRF-CEM, Murine colon 38, HCT-116, H-125, HepG2, CFU-GM	weak–mild	[43]
*Aspergillus wentii* EN-48	(1) 15 s 70% EtOH; (2) Rinsed in sterile water	*Sargassum* sp. (B)	asperolides A−C; tetranorditerpenoid derivative; wentilactones A-B; botryosphaerin B; LL-Z1271-*β*	terpenoids	SMMC-7721, HeLa HepG2, MCF-7 MDA-MB-231 NCI-H460, SW1990	10.00–17.00 µM	[44]
*Aspergillus wentii* EN-48	(1) 15 s 70% EtOH; (2) Rinsed in sterile water	*Sargassum* sp. (B)	wentilactone B	terpenoids	SMMC-7721, HepG2 Huh7, Hep3B	18.96 µM (SMMC-7721)	[45]
*Aspergillus wentii* EN-48	(1) 15 s 70% EtOH; (2) Rinsed in sterile water	*Sargassum* sp. (B)	wentilactone B	terpenoids	SMMC-7721	-	[46]
*Aspergillus wentii* EN-48	(1) 15 s 70% EtOH; (2) Rinsed in sterile water	*Sargassum* sp. (B)	wentilactone A	terpenoids	NCI-H460, NCI-H466	-	[47]
*Aspergillus wentii* EN-48	(1) 15 s 70% EtOH (2) Rinsed in sterile water	*Sargassum* sp. (B)	asperolide A	terpenoids	NCI-H460	-	[48]
*Aspergillus* sp. XNM-4	-	*Leathesia nana* (B)	(hydroxy(phenyl)methyl)-4H-pyran-4-one; 2-benzyl-4*H*-pyran-4-one; asperpyrone D; asperpyrone C; aurosperone B; fonsecinone B; asperpyrone B; dianhydro-aurasperone C; isoaurasperone A; aurasperone F; fonsecinone D; asperpyroneA; fonsecinone A; fonsecin; TMC 256 A1; flavasperone; carbonarone A; pestalamide A	polyketides	PANC-1, A549 MDA-MB-231, Caco-2, SK-OV-3	8.25 µM - potent on all cells	[49]
*Chaetomium globosum* QEN-14	(1) 15 s 70% EtOH; (2) Rinsed with sterile water	*Ulva pertusa*(G)	cytoglobosins A-G; isochaetoglobosin D; chaetoglobosin F_ex_	alkaloids	A-549, P388, KB	2.26–10.00 µM	[50]
*Cladosporium cladosporioides* EN-399	(1) 15 s 70% EtOH; (2) Rinsed with sterile water	*Laurencia okamurai* (R)	cladosporols F−I; cladosporol C; cladosporol J	polyketides	A549, Huh7, LM3, H446	1.00–5.00 µM	[51]
*Coniothyrium cereale*	(1) Rinsed with sterile H_2_O (3x); (2) 15 s 70% EtOH; (3) Rinsed in sterile artificial seawater (ASW)	*Enteromorpha* sp. (G)	coniosclerodin; (*Z*)-coniosclerodinol; (*E*)- coniosclerodinol; (15*S*, 17*S*)-(-)-sclerodinol; conioscleroderolide; coniosclerodione; coniolactone; (-)-7,8-dihydro-3,6-dihydroxy- 1,7,7,8-tetramethyl-5*H*-furo-[2’,3’:5,6]naphtho [1,8-bc]furan-5-one; (-)-sclerodin A; lamellicolic anhydride; (-) scleroderolide; (-) sclerodione	polyketides	HTB-9, HLE	7.20–20.00 µM	[52]
*Gibberella zeae*	-	*Codium fragile* (G)	3-hydroxy-5-(hydroxymethyl)-4-(4’-hydroxy phenoxy)pyrrolidin-2-one; (22*E*,24*R*)-7*β*,8*β*-epoxy-3*β*,5*α*,9*α*-trihydroxy ergosta-22-en-6-one;(22*E*,24*R*)-3*β*,5*α*,9*α*-trihy droxyergosta-7,22-dien-6-one; (22*E*,24*R*)-3*β*, 5*α*-dihydroxyergosta-7,22-dien-6-one; (22*E*,24*R*)-ergosta-7,22-dien-3*β*,5*α*,6*β*-triol; (22*E*,24*R*)-ergosta-5,22-dien-3*β*-ol; (22*E*,24*R*)- 5*α*,8*α*-epidioxyergosta-6,22-dien-3*β*-ol; (22*E*,24*R*)-5*α*,8*α*-epidioxyergosta-6,9(11),22-trien-3*β*-ol; (22*E*,24*R*)-1(10→6)- *abeo*ergosta-5,7,9,22-tetraen-3*α*-ol	alkaloids steroids	A-549 BEL-7402	17.60–61.80%	[53]
*Guignardia* sp.	(1) Washed with running tap water; (2) 1 min. 75% EtOH; (3) 5 min. 2.5% NaOCl; (4) Rinsed with sterile H_2_O (3x)	*Undaria pinnatifida* (B)	6, 22-diene-5, 8-epidioxyergosta-3-ol; ergosterol; cyclo-(Tyr-Leu); cyclo-(Phe-Phe); cyclo-(Val-Leu); cyclo-(Phe-Pro); cyclo-(Leu-Ile)	steroids peptides	KB	10.00–50.00 µg mL^−1^,	[54]
*Paecilomyces variotii* EN-291	(1) 15 s 70% EtOH (2) Rinsed in sterile water	Not identified (R)	dihydrocarneamide A; iso-notoamide B	alkaloids	NCI-H460	55.90–69.30 µmol L^−1^	[55]
*Paecilomycesvariotii* EN-291	(1) 15 s 70% EtOH (2) Rinsed in sterile water	*Grateloupia turuturu* (R)	varioloid A; varioloid B	alkaloids	A549, HCT116, HepG2	2.50–8.20 µg mL^−1^	[56]
*Penicillium chrysogenum* QEN-24S	(1) 15 s 70% EtOH (2) Rinsed in sterile water.	*Laurencia* sp. (R)	penicitides A-B; 2-(2,4-dihydroxy-6-methyl benzoyl)-glycerol; (2,4-dihydroxy-6-methyl benzoyl)-glycerol; penicimonoterpene	polyketides terpenoid	HepG2, NCI-H460, SMMC-7721, SW1990, DU145, Hela, MCF-7	32.00–40.00 µg mL^−1^	[57]
*Penicillium chrysogenum* QEN-24S	-	*Laurencia* sp. (R)	penicisteroids A-B; anicequol; (22*E*, 24*R*)-ergosta-4,6,8(14),22-tetraen-3-one; (22*E*, 24*R*)-ergosta-7,22-dien-3,6-dione; (22*E*, 24*R*)-5α,8α-epidioxyergosta-6,22-dien-3*β*-ol; (22*E*, 24*R*)-ergosta-5α,6α-epoxide-8, 22-dien-3*β*,7*α*-diol; (22*E*, 24*R*)-ergosta-7,22-dien-3*β*,5*α*,6*β*-triol; (22*E*, 24*R*)-ergosta-7,22-dien-3*β*,6*β*-diol	steroids	HeLa SW1990 NCI-H460	15.00–40.00 µg mL^−1^	[58]
*Penicillium* sp.	(1) 15 s 70% EtOH; (2) Rinsed in sterile water	*Ulva* sp. (G)	chromanone A	chromone	Hepa1c1c7, Cyp1A	4.00 µg mL^−1^	[59]
*Phoma* sp. NTOU4195	-	*Pterocladiella capillacea* (R)	phomaketides A-E; pseurotins A_3_ and G; FR-111142, pseurotins A, A_1_, A_2_, D and F_2_, 14-norpseurotin A; A-carbonylcarbene; tyrosol; cyclo(-_L_-Pro-_L_-Leu); cyclo(-_L_-Pro-_L_-Phe)	polyketides alkaloids phenylethanoid peptides	Endothelial progenitor cells (EPCs)	8.10–19.20 µM	[60]
*Talaromyce sislandicus* EN-501	(1) 15 s 70% EtOH (2) Rinsed in sterile water	*Laurencia* *okamurai* (R)	8-hydroxyconiothyrinone B; 8,11-dihydroxyconiothyrinone; 4*R*,8-dihydroxyconiothyrione B; 4*S*,8-dihydroxyconiothyrinone B; 4*S*,8-dihydroxy-10-*O*-methyldendryol E	quinines	A2780, A2780 CisR	<10.00 µM	[61]

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
