# Peer review of "Antitumor Potential of Seaweed Derived-Endophytic Fungi"

_antibiotics, 2019, doi:10.3390/antibiotics8040205_

Round 1

Reviewer 1 Report

A general observation is that the manuscript is considered of good quality and scientifically sound, containing amble information and summarizing the relevant current knowledge of the past decade in both table and figure format. There is a need for improvements which will make the manuscript more comprehensive and reader friendly. These improvements are mainly focused on the use of English, as there are numerous grammatical and syntax errors that need to be corrected, but also on the issue of potential side-effects of these natural products. The comments provided on the use of English are only indicative and the list of necessary corrections is not exhaustive, so the authors are kindly advised to have their manuscript thoroughly checked by a native English speaker or a relevant professional service.

Abstact:
- Page 1, line 18: Please omit “antiviral”, it is mentioned twice.
- Page 1, line 18: Please correct “genotoxicity” to “genotoxic”.
- Page 1, lines 19-20: “Among these activities, anticancer still worries...” please rephrase, it does not make sense.
- Page 1, line 24: “isolated of” change to “isolated from”.
- Page 1, line 25: “describing” change to “it describes”.

1. Introduction:
- Page 1, line 33: Please correct “metastasis” to “metastases” (it is plural as described).
- Page 2, lines 45-47: “...and, they source come from plants, microorganisms (marine and terrestrial) and marine organisms, many of them are potential anticancer agents”- please rephrase, it does not make sense.
- Page 2, lines 47-50: please rephrase, it does not make sense – confusing syntax.
- Page 2, lines 58-62: please rephrase, it does not make sense – confusing syntax.
- Page 2, line 63: Please correct “The microorganism’s classes” to “The microorganisms’ classes”.
- Page 2, line 69: Please omit the second “antiviral”, it is mentioned twice.
- Page 2, line 70: Please correct “genotoxicity” to “genotoxic”.
- Page 2, line 78: Please correct “which have been showed” to “which have shown”.
- Page 2, line 81: “In this work, it will be related the compounds”- please rephrase, it does not make sense – confusing syntax.
- Page 2, line 83: Please correct “showed” to “shown”.

- Page 2, lines 86-89: Are natural antitumor compounds completely free of side-effects for cancer patients? For example, Halaven® is reported to have numerous side effects for patients, e.g. “HALAVEN can cause changes in your heartbeat (called QT prolongation). This can cause irregular heartbeats. Your health care provider may do heart monitoring (electrocardiogram or ECG) or blood tests during your treatment with HALAVEN to check for heart problems.
The most common side effects of HALAVEN in adults with breast cancer include low white blood cell count (neutropenia), low red blood cell count (anemia), weakness or tiredness, hair loss (alopecia), nausea, and constipation.”
https://www.halaven.com/metastatic-breast-cancer
Please take into account these facts and comment/correct throughout the text all definite expressions on the lack of side effects of natural products. It could be possible that they may have fewer implications for human health, but it is certain that they are not completely free of adverse effects (clinical trials have shown this), so the text should reflect the real situation and written cautiously on this matter.

2. Endophytic fungi
- Page 3, line 118: Please correct “the ability of” to “the ability to”.

3. Cytotoxic secondary metabolites produced by endophytic fungi
- Page 9, line 214 (first instance): please indicate that the numbering of the compounds refers to Figure 1 for structures – there is no reference to Figure 1 in all these sections.
- Page 11, line 295: please change numbering to 3.4.
- Page 11, line 296: please change numbering to 3.4.1.
- Page 12, line 342: please change numbering to 3.4.2.

4. Biotechnology of marine endophytic fungi
- Page 16, line 452: please avoid using “great”, it would be better to use “promising”.
- Page 17, lines 504-505: “Could be cited these strategies”: please rephrase the sentence, the syntax is confusing.

Author Response

Dear  Molly Li

Assistant Editor,

Thank you so much for the carefully revision concerning to our manuscript (#614353) submitted to journal Antibiotics. All reviewers' suggestions were considered and corrected throughout the marked manuscript. Other changes are also highlighted in the text.

In this way, we hope that all of the corrections was done and it is in accordance to the Journal quality. However, we are able to answer any additional question suggested by the reviewers or editor.

Sincerely,

Hosana Maria Debonsi

Reviewer 1:

A general observation is that the manuscript is considered of good quality and scientifically sound, containing amble information and summarizing the relevant current knowledge of the past decade in both table and figure format. There is a need for improvements which will make the manuscript more comprehensive and reader friendly. These improvements are mainly focused on the use of English, as there are numerous grammatical and syntax errors that need to be corrected, but also on the issue of potential side-effects of these natural products. The comments provided on the use of English are only indicative and the list of necessary corrections is not exhaustive, so the authors are kindly advised to have their manuscript thoroughly checked by a native English speaker or a relevant professional service.

Reply: The entire grammar and syntax of the text was reviewed by the authors of the manuscript as suggested by reviewer.

Abstract

- Page 1, line 18: Please omit “antiviral”, it is mentioned twice.

Reply: Revised in page 1, line 18.

- Page 1, line 18: Please correct “genotoxicity” to “genotoxic”.

Reply: Revised in page 1, line 18.

- Page 1, lines 19-20: “Among these activities, anticancer still worries...” please rephrase, it does not make sense.

Reply: Revised in page 1, lines 19-22.

- Page 1, line 24: “isolated of” change to “isolated from”.

Reply: Revised in page 1, line 24.

- Page 1, line 25: “describing” change to “it describes”.

Reply: Revised in page 1, line 25.

Introduction:

- Page 1, line 33: Please correct “metastasis” to “metastases” (it is plural as described).

Reply: Revised in page 1, line 33.

- Page 2, lines 45-47: “...and, they source come from plants, microorganisms (marine

and terrestrial) and marine organisms, many of them are potential anticancer agents”

please rephrase, it does not make sense.

Reply: Revised in page 2, lines 46-48.

- Page 2, lines 47-50: please rephrase, it does not make sense – confusing syntax.

Reply: Revised in page 2, lines 48-51.

- Page 2, lines 58-62: please rephrase, it does not make sense – confusing syntax.

Reply: Revised in page 2, lines 59-62.

- Page 2, line 63: Please correct “The microorganism’s classes” to “The microorganisms’

classes”.

Reply: Revised in page 2, line 63.

- Page 2, line 69: Please omit the second “antiviral”, it is mentioned twice.

Reply: Revised in page 2, line 69.

- Page 2, line 70: Please correct “genotoxicity” to “genotoxic”.

Reply: Revised in page 2, line 70.

- Page 2, line 78: Please correct “which have been showed” to “which have shown”.

Reply: Revised in page 2, line 79.

- Page 2, line 81: “In this work, it will be related the compounds”- please rephrase, it does not make sense – confusing syntax.

Reply: Revised in page 2, line 82.

- Page 2, line 83: Please correct “showed” to “shown”.

Reply: Revised in page 2, line 84.

- Page 2, lines 86-89: Are natural antitumor compounds completely free of side-effects for cancer patients? For example, Halaven® is reported to have numerous side effects for patients, e.g. “HALAVEN can cause changes in your heartbeat (called QT prolongation). This can cause irregular heartbeats. Your health care provider may do heart monitoring (electrocardiogram or ECG) or blood tests during your treatment with HALAVEN to check for heart problems. The most common side effects of HALAVEN in adults with breast cancer include low white blood cell count (neutropenia), low red blood cell count (anemia), weakness or tiredness, hair loss (alopecia), nausea, and constipation.” https://www.halaven.com/metastatic-breast-cancer.

Please take into account these facts and comment/correct throughout the text all definite expressions on the lack of side effects of natural products. It could be possible that they may have fewer implications for human health, but it is certain that they are not completely free of adverse effects (clinical trials have shown this), so the text should reflect the real situation and written cautiously on this matter.

Replay: The reviewer's considerations regarding adverse drug effects mentioned in the introduction have been revised and corrected throughout the text in page 1 lines 42-44 and page 2 lines 87-90.

Endophytic fungi

- Page 3, line 118: Please correct “the ability of” to “the ability to”.

Reply: Revised in page 3, line 120.

Cytotoxic secondary metabolites produced by endophytic fungi

- Page 9, line 214 (first instance): please indicate that the numbering of the compounds refers to Figure 1 for structures – there is no reference to Figure 1 in all these sections.

Reply: Revised in all sections.

- Page 11, line 295: please change numbering to 3.4.

Reply: Revised in page 11, line 300.

- Page 11, line 296: please change numbering to 3.4.1.

Reply: Revised in page 11, line 301.

- Page 12, line 342: please change numbering to 3.4.2.

Reply: Revised in page 12, line 348.

Biotechnology of marine endophytic fungi

- Page 16, line 452: please avoid using “great”, it would be better to use “promising”.

Reply: Revised in page 16, line 458.

- Page 17, lines 504-505: “Could be cited these strategies”: please rephrase the sentence, the syntax is confusing.

Reply: Revised in page 17, lines 510-511.

Reviewer 2 Report

The manuscript by Terxeira et al. comprehensively reviewed the current proceedings in the field of antitumor drug discovery from the seaweed derived-endophytic fungi. The review is very well written and organized. I believe this review will sufficiently inform the readers on the advances in this field. Only one minor question is that why do seaweed endophytic fungi stand out among other organisms, like marine algae or bacteria? It may help readers understand the significance in the introduction. Some minor issues are listed below,

L32, typo. Some types of…

L60-62, this sentence reads awkward to me. Please consider refrasing.

L78, remove been

L84, in vitro and/or in vivo

Author Response

Dear  Molly Li

Assistant Editor,

Thank you so much for the carefully revision concerning to our manuscript (#614353) submitted to journal Antibiotics. All reviewers' suggestions were considered and corrected throughout the marked manuscript. Other changes are also highlighted in the text.

In this way, we hope that all of the corrections was done and it is in accordance to the Journal quality. However, we are able to answer any additional question suggested by the reviewers or editor.

Sincerely,

Hosana Maria Debonsi

Reviewer 2:

The manuscript by Terxeira et al. comprehensively reviewed the current proceedings in the field of antitumor drug discovery from the seaweed derived-endophytic fungi. The review is very well written and organized. I believe this review will sufficiently inform the readers on the advances in this field. Only one minor question is that why do seaweed endophytic fungi stand out among other organisms, like marine algae or bacteria? It may help readers understand the significance in the introduction. Some minor issues are listed below:

Reply: Endophytic fungi of marine organisms stand out mainly for the ability to live within tissues without causing apparent damage to the host, and establish a relationship that may be beneficial to both. Marine endophytic fungi have physiological and ecological roles for the fungal-host interaction, which include nutritional enhancement, stabilization of the host skeleton, and production of various metabolites. The production of these metabolites provides chemical adaptation to environmental conditions, substrate competition, helps protect the host against pathogen attacks and can also be evaluated for bioactive capacity. We chose endophytic seaweed fungi mainly because it is an object of study of our research group and we wish to study the subject further.

L32, typo. Some types of…

Reply: Revised in page 1, line 32.

L60-62, this sentence reads awkward to me. Please consider rephrasing.

Reply: Revised in page 2, lines 59-62.

L78, remove been

Reply: Revised in page 2, line 79.

L84, in vitro and/or in vivo

Reply: Revised in page 2, line 85.